# Phylogenetic Species of *Paracoccidioides* spp. Isolated from Clinical and Environmental Samples in a Hyperendemic Area of Paracoccidioidomycosis in Southeastern Brazil

**DOI:** 10.3390/jof6030132

**Published:** 2020-08-11

**Authors:** Tiago A. Cocio, Erika Nascimento, Marcia R. von Zeska Kress, Eduardo Bagagli, Roberto Martinez

**Affiliations:** 1Department of Internal Medicine, Ribeirão Preto Medical School, University of São Paulo, São Paulo, Ribeirão Preto 14049-900, Brazil; erika.nascimento@gmail.com (E.N.); rmartine@fmrp.usp.br (R.M.); 2Department of Clinical Analyses, Toxicology and Food Science, School of Pharmaceutical Sciences of Ribeirão Preto, University of São Paulo, São Paulo, Ribeirão Preto 14040-903, Brazil; kress@fcfrp.usp.br; 3Department of Chemical and Biological Sciences, Institute of Biosciences—UNESP, São Paulo, Botucatu 18618-691, Brazil; eduardo.bagagli@unesp.br

**Keywords:** *Paracoccidoides brasiliensis sensu stricto* (S1a and S1b), *Paracoccidioides americana* (PS2), *Paracoccidioides restrepiensis* (PS3), *Paracoccidoides lutzii*, Paracoccidioidomycosis, eco-epidemiology

## Abstract

*Paracoccidioides brasiliensis* complex and *P. lutzii* are the etiological agents of paracoccidioidomycosis. The geographic distribution of these species in South America is still poorly comprehended. Fifty samples of *Paracoccidioides* spp. were genotyped, with 46 clinical isolates predominantly isolated in the geographic area of Ribeirão Preto, SP, and four environmental isolates collected in Ibiá, MG, southeastern Brazil. These isolates were evaluated by PCR-RFLP (Restriction Fragment Length Polymorphism) of the *tub*1 gene and the sequencing of the *gp43* exon 2 loci. The species *P. lutzii* was confirmed by sequencing the internal transcribed spacer (ITS) region of the ribosomal DNA. *P. brasiliensis sensu stricto* S1b (*n* = 42) and S1a (*n* = 5), *P. americana* (*n* = 1), *P. restrepiensis* (*n* = 1), and *P. lutzii* (*n* = 1) were identified among the clinical isolates. All the environmental isolates were characterized as *P. brasiliensis sensu stricto* S1b. The patient infection by *P. lutzii*, *P. americana* (PS2), and one isolate of *P. brasiliensis sensu stricto* S1b most likely occurred in a geographic area far from the fungal isolation site. No association was found between the infecting genotype and the disease form. These results expand the knowledge of the *Paracoccidioides* species distribution and emphasize that human migration must also be considered to pinpoint the genotypes in the endemic area.

## 1. Introduction

Paracoccidioidomycosis (PCM) is a systemic fungal infection endemic to Latin America. Reports of PCM in South American countries show that about 80% of the cases are from Brazil, mainly in the southeast, south, and midwest regions, besides a low number of cases in the north and northeast regions [1].

PCM is caused by thermodimorphic fungi belonging to the genus *Paracoccidioides*, family *Ajellomycetaceae*, order *Onygenales*, class *Eurotiomycetes*, and is represented by five distinct species, including *P. brasiliensis sensu stricto* (variants S1a and S1b), which has been isolated in Brazil, Argentina, Paraguay, Peru, and Venezuela; *P. americana* (also known as PS2), distributed in Brazil and Venezuela; *P. restrepiensis* (also known as PS3), found mainly in Colombia; *P. venezuelensis* (also known as PS4), exclusively found in Venezuela; and *P. lutzii*, most commonly isolated in the midwest region of Brazil [2,3,4,5].

Determining the geographic distribution of the species of the genus *Paracoccidioides* spp. requires studies with epidemiological and clinical data, since important factors such as regional migration and the long latency period of PCM can make it difficult to know the exact location of the infection and the occurrence of each genotype in the endemic regions of PCM [6,7]. Genotypic studies have indicated that the predominant causative species of PCM in Brazil is *P. brasiliensis sensu stricto* (S1a and S1b). Its wide geographic occurrence includes particularly the southeast and south regions of Brazil (the states of Minas Gerais, São Paulo, Rio de Janeiro, Espírito Santo, Paraná, and Rio Grande do Sul) [8,9,10]. On a smaller scale, there are reports of the occurrences of *P. americana* (PS2) and *P. restrepiensis* (PS3) (southeast and south regions) in Brazil [8,11,12]. Brazil also has the largest endemic area of *P. lutzii* (midwest and north regions) [13,14].

Patients usually have the chronic PCM form, but it has not yet been determined whether certain genotypes may be related to clinical manifestations of this fungal disease [7].

This study aimed to evaluate genotypically the isolates of *Paracoccidioides* spp. from the Ribeirão Preto region, located in northeastern São Paulo state, southeast of Brazil, and relate the species to the clinical and epidemiological data of patients. The genotypic identification of *Paracoccidioides* spp. recovered from PCM cases in a hyperendemic area of the southeast of Brazil is unprecedented and may contribute to the eco-epidemiology of this mycosis.

## 2. Materials and Methods

### 2.1. Samples of Paracoccidioides spp. and Cultivation Conditions

Forty four samples of *Paracoccidioides* spp. isolated from patients treated at the Hospital das Clínicas of Ribeirão Preto Medical School, University of São Paulo—FMRP/USP (Ribeirão Preto, SP, Brazil)—from 1975 to 2019; two samples isolated from patients from Foz do Iguaçu, Paraná state (Appendix A); and four environmental samples (one soil and three armadillos isolates) collected in Ibiá, Minas Gerais state, were used in this study (Appendix A) [15,16]. The reference isolates were Pb18 (collected from the Laboratory of Medical Mycology Research of the Ribeirão Preto Medical School), representative of the species *P. brasiliensis sensu stricto* (S1b) [5]; Pbdog-EPM 194, representative of the species *P. americana* (PS2) [17]; T2-EPM 54, representative of the species *P. restrepiensis* (PS3) [18]; and Pb01, representative of the species *P. lutzii* [4] (Appendix A). These isolates were maintained by successive subcultures on Sabouraud dextrose agar (Oxoid^®^ LTD—Thermo Fisher Scientific^®^, Basingstoke, Hampshire, UK), plus 0.15 g/L of chloramphenicol sodium succinate (Blau Farmacêutica, Florianópolis, SC, Brazil), and incubated at 25 °C.

### 2.2. Genomic DNA Extraction from Paracoccidioides spp.

The genomic DNA of the isolates was obtained from mycelia grown in the modified synthetic liquid culture medium McVeigh and Morton [19] for 35 days at 25 °C in an orbital stirrer (Infors HT—Ecotron^®^, Bottmingen, Switzerland) at 130 rpm. The genomic DNA extraction was performed according to the method I (treated glass beads/phenol-chloroform/isoamyl alcohol) described by van Burik in 1998, with minimal modifications [20].

After obtaining the genomic DNA from the clinical and environmental isolates, treatment with RNase A^®^ (Thermo Fisher Scientific Inc.^®^, Waltham, MA, USA) was carried out following the manufacturer’s standards. The concentrations (ng/µL) of genomic DNA from the samples were determined using NanoDrop 2000^®^ (Thermo Fisher Scientific Inc.^®^, Waltham, MA, USA).

### 2.3. Methods and Primers Used in the Molecular Characterization of Paracoccidioides spp.

The molecular characterization of clinical isolates was determined using the PCR-RFLP (Restriction Fragment Length Polymorphism) techniques of the *tub*1 gene [18] and the sequencing of the *gp43* exon 2 loci gene, which encodes the 43 kDa glycoprotein [10], to identify the species and varieties belonging to the genus *Paracoccidioides*. The isolate not identified by the PCR-RFLP technique of the *tub*1 gene was evaluated by sequencing the internal transcribed spacer (ITS) region of the ribosomal DNA to confirm the species of the genus *Paracoccidioides* [10,21]. The primers used in both methods, the annealing temperature, and their references are described in Appendix A. The PCR reactions for each technique performed in this study were carried out using the vapo.protect^®^ thermocycler (Eppendorf, Hamburg, Germany) and the Taq polymerase enzyme—GoTaq^®^ Green Master Mix (Promega, Madison, WI, USA)—according to the manufacturer’s instructions.

The molecular characterization using the PCR-RFLP technique generated PCR *tub*1 products approximately 263 bp in size, being subjected to double digestion with the *Bcl*I and *Msp*I endonucleases (Thermo Fisher Scientific Inc.^®^, Waltham, MA, USA) at a concentration of 10 U/µL, each for 16 h at 37 °C, according to the manufacturer’s instructions. The fragments generated after double enzymatic digestion were submitted to electrophoresis on 3% agarose gel at 70 V for 180 min [18], and visualized in the presence of the SYBR^®^ Safe DNA gel stain (Thermo Fisher Scientific Inc.^®^, Waltham, MA, USA) with a 50 bp DNA ladder molecular weight marker (Sinapse^®^ Inc., United States). The fragments generated after enzymatic digestion by endonucleases were visualized on the ChemoDoc™ XRS+ photo-documenter and photographed using the Image Lab™ software (Bio-Rad^®^, Hercules, CA, USA).

### 2.4. Sequencing of ITS1-5.8S-ITS2 rDNA Region, gp43 Exon 2 Loci, and Phylogenetic Analysis

For the sequencing of the ITS1-5.8S-ITS2 rDNA region and the *gp*43 exon 2 loci [10,21], the PCR products were purified using the Wizard^®^ SV Gel kit and PCR Clean-UP System (Promega, Madison, WI, USA), according to the manufacturer’s instructions. The nucleotide sequences were determined using the platform of the Human Genome and Stem-Cell Research Center of the Institute of Biosciences of the University of São Paulo (CEGH-CEL/IB-USP) with the ABI3730 DNA Analyzer^®^ sequencer (Applied Biosystems, Foster City, CA, USA), using the BigDye^®^ Terminator v3.1 Cycle Sequencing Kit (Thermo Fisher Scientific Inc.^®^, Waltham, MA, USA), according to the manufacturer’s instructions. The nucleotide sequences of the ITS1-5.8S-ITS2 rDNA region and the *gp43* exon 2 loci were analyzed using the ChromasPro^®^ software (ChromasPro^®^ 2.6.5, Technelysium Pty Ltd., Tewantin, QLD, Australia). The nucleotide sequences were compared with the database using the BLASTn—Basic Local Alignment Search Tool: https://blast.ncbi.nlm.nih.gov/Blast.cgi [22]. 

A phylogenetic analysis of clinical isolates was performed using the maximum likelihood (ML) method, in which the Tamura-Nei evolutionary model was applied to the ITS1-5.8S-ITS2 rDNA region sequences and the Kimura 2-parameter model to the *gp43* exon 2 loci sequences, both with a 1000 replication bootstrap [10]. The multiple alignments using the Clustal W tool and the phylogenetic reconstruction were performed with the MEGA 6.0 software (Molecular Evolutionary Genetics Analysis) [23]. The *Histoplasma capsulatum* AMC_HC002 sequence was included in the phylogenetic analysis of the ITS1-5.8S-ITS2 rDNA region under the accession number KT275850.1 as an external group, and the B1-T1F1 nucleotide sequence was inserted to analyze the *gp43* exon 2 loci as a reference for the S1a variety of *P. brasiliensis sensu stricto* under the GenBank accession number DQ003724.1 [10].

The nucleotide sequences of the studied isolates were deposited in the GenBank under the accession numbers of MK909758-MK909806 for *gp43* exon 2 loci and MK886790 for ITS1-5.8S-ITS2 rDNA—HCRP 191 (Appendix A).

### 2.5. Epidemiological Data of Patients with Isolation of Paracoccidioides spp.

The medical records of patients were analyzed to obtain information regarding the PCM clinical form and the geographic origin of each studied individual. The epidemiological data, residence, and the clinical manifestation of the patient were related to the genotypic data of the clinical isolates of *Paracoccidioides* spp. obtained in this study.

### 2.6. Ethics

The study was approved by the Research Ethics Committee of the Hospital das Clínicas of Ribeirão Preto Medical School, University of São Paulo—FMRP/USP (Protocol HCRP 4456/2017), 17/04/2017.

## 3. Results

### 3.1. Molecular Identification of Paracoccidioides spp. Isolates by the PCR-RFLP Technique of the tub1 Gene

Forty seven (including environmental isolates—Appendix A) among the 50 isolates analyzed in this study by the PCR-RFLP technique of the *tub*1 gene (46 clinical isolates and 4 environmental isolates—Appendix A) were identified as belonging to the species *P. brasiliensis sensu stricto* (S1) because the *Bcl*I enzyme recognized the restriction site, generating two fragments (155 and 108 bp). The *tub*1 gene of isolate HCRP 199 had its cleavage site recognized by both enzymes (*Bcl*I and *Msp*I), generating three fragments (62, 93, and 108 bp) and being classified as *P. americana* (PS2). The HCRP BAT isolate was identified as *P. restrepiensis* (PS3), because the endonucleases did not recognize the *tub*1 cleavage site, the gene remaining in its total integrity (263 bp) (Figure 1). The *tub*1 gene of the HCRP 191 isolate was not amplified by conventional PCR, and therefore the determination of the phylogenetic species of *Paracoccidioides* spp. by PCR-RFLP was inconclusive (Appendix A).

### 3.2. Genotypic Identification and Phylogenetic Analysis Using the gp43 Exon 2 loci Gene Sequencing

Among the 50 strains analyzed by the *gp43* exon 2 loci sequencing, 47 isolates (including environmental samples) were identified as *P. brasiliensis sensu stricto* (S1); 5 isolates belonging to the S1a variety, with a 100% identity to DQ003724.1 B1–T1F1 (S1a); and 42 isolates (38 clinical and 4 environmental strains) belonging to the S1b variety, with a 100% similarity to DQ003729.1 B17–Pb18 on GenBank. The isolate HCRP 199, characterized as *P. americana* (PS2), has a 100% identity to the standard sequence *P. americana* (PS2) DQ003736.1 B13–Uberlândia. The isolate HCRP BAT, identified as *P. restrepiensis* (PS3), has a 100% identity to the standard sequence *P. restrepiensis* (PS3) KT251008.1-T2–EPM54. The isolate HCRP 191 was genotypically characterized as *P. lutzii*, showing a 100% identity to EU870196.1 of Pb01 (*P. lutzii*), was deposited in GenBank (Appendix A).

The phylogenetic analysis of the *gp43* exon 2 loci sequences shows that 5 clinical isolates have an evolutionary proximity to DQ003724.1 B1–T1F1 *P. brasiliensis sensu stricto* (S1a), and 42 isolates are more similar to B17–Pb18 *P. brasiliensis sensu stricto* (S1b), including the environmental strains collected in Ibiá, MG, Brazil. The isolate HCRP199 is similar to EPM194–Pbdog *P. americana* (PS2), HCRP BAT has a genetic proximity to EPM54–T2 *P. restrepiensis* (PS3), and the isolate HCRP191 has a phylogenetic proximity to Pb01–*P. lutzii* (Figure 2).

### 3.3. The HCRP 191 Strain was Confirmed as P. lutzii by Sequencing the ITS1-5.8S-ITS2 rDNA Region

The sequencing of the ITS1-5.8S-ITS2 rDNA region was performed to validate the identification of the species of the genus *Paracoccidioides*, as the isolate HCRP 191 was not molecularly identified by the PCR-RFLP technique of the *tub*1 gene. A 99.50% identity was observed with the ITS nucleotide sequence of Pb01 (*P. lutzii*) (EU870297.1), confirming the isolate as belonging to the species *P. lutzii* (Spreadsheet S1).

A phylogenetic analysis of the ITS1-5.8S-ITS2 rDNA region shows that the isolate HCRP191 has a similarity with Pb01 (*P. lutzii*) and a phylogenetic distance relative to Pb18 (S1b), EPM194–Pbdog (PS2), and EPM54—(PS3), which are species of the *Paracoccidioides* spp. complex and *H. capsulatum* AMC_HC002 (Figure 3).

### 3.4. Geographic Origin of Patients and Clinical form of PCM

Among the 46 isolates of clinical origin characterized genotypically in this study, 67.4% of the patients manifested the chronic form and 32.6% manifested the subacute form of PCM. Evaluating the clinical PCM manifestation by species of the *Paracoccidioides* spp. complex, 38 out of the 46 genotyped clinical isolates were characterized as *P. brasiliensis sensu stricto* variant S1b, in whose patients 65.8% presented the chronic form and 34.2% presented the subacute form. Among the five clinical isolates identified as *P. brasiliensis sensu stricto* variant S1a, 60% of the patients manifested the chronic form, and 40% the subacute form. Patients from isolates HCRP 191 (*P. lutzii*) and HCRP 199 (*P. americana*–PS2) presented the chronic form. HCRP BAT (*P. restrepiensis*–PS3) was isolated from a patient with the subacute form of the disease (Table 1).

The geographic origin showed that, among the 46 patients, 39 lived in the city and region of Ribeirão Preto, SP, Brazil. The patients from isolates HCRP 063 (*P. brasiliensis sensu stricto* S1b), HCRP 191 (*P. lutzii*), and HCRP 199 (*P. americana*–PS2) (Appendix A) migrated to Ribeirão Preto, SP, Brazil, from the States of Maranhão, Mato Grosso, and Paraná, respectively. Six patients acquired PCM while residing in other endemic areas in the southeast and south of Brazil (Minas Gerais State—4; and Paraná State—2) (Figure 4 and Table 1).

## 4. Discussion

The knowledge of the biogeography of the genus *Paracoccidioides* spp. is hampered by the migration of patients between the different PCM endemic regions, the long latency period of the chronic PCM form, and the difficulty in obtaining environmental isolates of this fungus [24]. This study evaluated genotypically *Paracoccidioides* spp. isolates from a single and defined endemic PCM area, partially covering the states of São Paulo and Minas Gerais in the southeast of Brazil [25]. Additionally, two isolates from patients in the region of Foz do Iguaçu, PR, in the south of Brazil, were also evaluated.

Among the clinical isolates genotypically characterized in this study, *P. brasiliensis sensu stricto* (S1a and S1b) showed a higher prevalence in the region of Ribeirão Preto, SP, Brazil, thus confirming reports in the literature on the high incidence of this phylogenetic species in South America and Brazil, particularly in the southeast and south [7,9,10]. The prevalent variety of *P. brasiliensis sensu stricto* among the clinical isolates evaluated in this study is S1b, which was identified in 38 (82.6%) out of 46 patients. Except for one case that apparently acquired the fungal infection in the Maranhão State, the other patients lived in the states of São Paulo, Minas Gerais, and Paraná. The S1a variant was identified in 5 out of 46 patients with geographic origin in cities in the region of Ribeirão Preto, SP, Brazil (Table 1). The combined geographic distribution of *P. brasiliensis sensu stricto* S1a and S1b shows that 40 out of 43 individuals have remained in the states of the southeast since birth or have moved from the contiguous states of Minas Gerais and Paraná to the region of Ribeirão Preto, SP, Brazil, where they expressed PCM. No relationship was observed between the geographic area of residence of patients and the genotypic variants S1a and or S1b of *P. brasiliensis sensu stricto*. The patient in one of the cases of PCM caused by *P. brasiliensis sensu stricto* S1b (isolate HCRP 170—Appendix A) had AIDS and several relapses of opportunistic PCM until death. A phylogenetic and population study using complete genomes from 31 clinical isolates of *Paracoccidioides* spp. from some countries in South America identified *P. brasiliensis sensu stricto* S1a as the predominant variant in Argentina and the S1b variant distributed between Paraguay and Argentina [26]. This study also showed the occurrence of genotypes with some degree of admixing between S1a and S1b [26]. Therefore, the differentiation and/or classification between genotypes S1a and S1b based only on the two molecular markers used here (RFLP standard of the *tub*1 gene and *gp43* exon 2 polymorphism) can lead to different results from those obtained by the analysis of the complete genome, but without changing the classification in the S1 group. Therefore, the differentiation of the variants of *P. brasiliensis sensu stricto* (S1a and S1b) carried out in this study has the limitation of the used methodology.

The phylogenetic species PS2 (*P. americana*) has been isolated with a lower prevalence of patients and environmental samples in endemic areas in the south and southeast of Brazil [10]. The isolate HCRP 199 (*P. americana*—PS2) was obtained from a 57-year-old male patient who presented the chronic form of PCM and lived in Ribeirão Preto, SP, although this individual had lived for a long time in Parana state (city of Londrina). *P. americana* (PS2) was also isolated from a dog with generalized lymphadenomegaly in Curitiba, PR [17]. Additionally, *P. americana* (PS2) was isolated from armadillos from rural regions of Botucatu, SP, showing that this species is endemic to the southeast of Brazil [10]. Recently, *P. americana* (PS2) was isolated in Rio de Janeiro state from patients with the chronic form of PCM [8]. Moreover, *P. americana* (PS2) was also isolated from patients with PCM in other Latin American countries, such as Venezuela, Uruguay, and Argentina [18].

*P. restrepiensis* (PS3) is another species that has been identified in endemic areas of Brazil but has its geographic distribution centered in Colombia. The isolate HCRP BAT, identified as the PS3 genotype, was obtained from a 33-year-old male patient who presented the subacute form of PCM. The patient, who was born and lived in the region of Ribeirão Preto, SP, had no history of travel to other geographic regions of the Brazilian territory and countries of Latin America, suggesting the acquisition of the fungus in that region of the southeast of Brazil [11]. The isolate HCRP BAT (also called Pb 327-B) had been evaluated together with *P. brasiliensis* samples from different South American countries using techniques such as RAPD and RFLP, being grouped close to *P. brasiliensis* (PS3) isolates from Colombia [27,28]. Recently, an isolate of a patient from Botucatu, in the São Paulo State, was identified as *P. restrepiensis* (PS3) [12], a genotype also isolated in Argentina and Peru [26].

*P. lutzii* has a predominant geographic distribution in the midwest region of Brazil, in the states of Mato Grosso, Mato Grosso do Sul, and Goiás, and its isolation is not usual in the southeast of Brazil [9,29]. The isolate HCRP 191 (*P. lutzii*) was obtained from a 55-year-old male patient who presented the chronic form of PCM. He was a resident of Ribeirão Preto, SP, Brazil, but was born in Parana state, in the south region, and had a long history of work and residence in rural areas of municipalities in the midwest region, where he probably acquired the fungal infection, which appeared as PCM years later in the southeast region of Brazil. However, *P. lutzii* can live saprophytically in the southeast and south of Brazil, as the isolate called IFM 54648 (or LDR 2), genotypically similar to Pb01 (*P. lutzii*), was obtained from a patient residing in Londrina, PR, but who had also resided in Botucatu, SP [30]. In addition, molecular traces of *P. lutzii* were found in soil and aerosol samples from armadillo burrows in the southeast of Brazil (São Paulo and Minas Gerais) [6,31], suggesting that this species is not found exclusively in the midwest of Brazil.

Most cases of PCM have a chronic disease form [7]. Besides the geographic origin and genotypic data, the clinical manifestation of PCM is an important characteristic to understand the pathogenicity of species of *Paracoccidioides* spp. complex and *P. lutzii*. Recent studies have evaluated the association of etiological agents (*P. brasiliensis sensu stricto*, *P. americana*, and *P. lutzii*) in the southeast and midwest regions of Brazil with the respective clinical form and found that most patients had the chronic disease form, regardless of the species of the genus *Paracoccidioides* [6,8,29]. In this study, the chronic form of PCM was observed in most cases attributed to *P. brasiliensis sensu stricto* and in patients infected with *P. americana* (PS2) and *P. lutzii*. The patient infected with *P. restrepiensis* (PS3) had the subacute form of PCM, but most of the clinical isolates PS3 correspond to cases of the chronic form [11,12]. It has not been possible to associate phylogenetic species with the clinical form of PCM, as the chronic form of PCM has been predominant in this and other studies in cases related to different *Paracoccidioides* spp. genotypes [7,29].

The environmental isolates evaluated here were collected approximately 20 years ago in Ibiá, Minas Gerais state, and have been kept since then in culture collection. The Ibiá isolate comes from the soil of a coffee plantation, and the other three isolates (Ibiá T1, Ibiá T2, and Ibiá T3) were obtained from armadillos (*Dasypus novemcinctus*) captured in the same municipality, whose territory is close to the area where some patients live with PCM [15,16]. Ibiá T1 (called EPM 101) was previously identified as *P. brasiliensis sensu stricto* by the PCR-RFLP method of the *tub*1 gene [18]. A phylogenetic study that includes soil and armadillo samples showed *P. brasiliensis sensu stricto* S1a and S1b as the main agents of human PCM and in environmental samples from southeast Brazil [10].

The obtained data contribute to the knowledge of speciation related to *Paracoccidioides* spp. and show that human migration should be considered in the biogeography of the genus *Paracoccidioides*.

## Figures and Tables

**Figure 1 jof-06-00132-f001:**
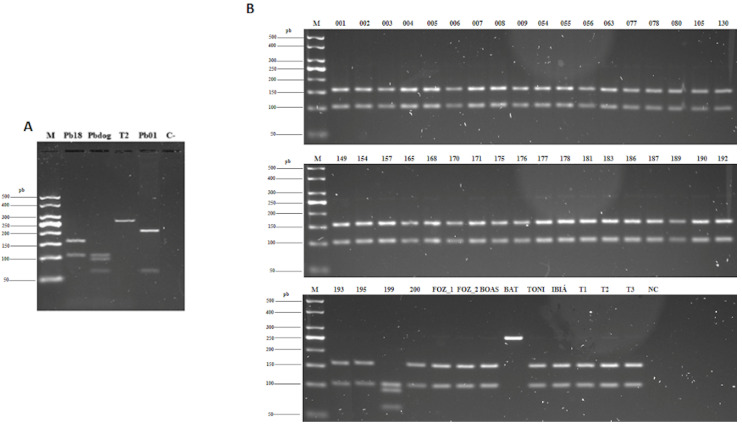
Patterns of fragments obtained after digestion with endonucleases *Bcl*I and *Msp*I (PCR-RFLP of *tub1* gene), showing a similarity between the clinical and environmental isolates of *Paracoccidioides* spp. and reference strains. (**A**) 3% agarose gel with reference strains: Pb18—*P. brasiliensis sensu stricto* (S1b); Pbdog—EPM 194—*P. americana* (PS2); T2—EPM 54—*P. restrepiensis* (PS3); Pb_01—*P. lutzii*. (**B**) 3% agarose gel with clinical (001 to FOZ_2, BAT, and TONI) and environmental (Ibiá, T1, T2, and T3) isolates evaluated in this study. Isolates 199 and BAT show a pattern corresponding, respectively, to Pbdog (*P. americana* (PS2)) and T2 (*P. restrepiensis* (PS3)), while the others have a pattern comparable to Pb18 (S1). The isolate HCRP 191 was not evaluated because there was no amplification of the *tub1* gene. **M:** 50-bp DNA ladder molecular weight marker (Sinapse^®^ Inc., USA). **NC:** negative control.

**Figure 2 jof-06-00132-f002:**
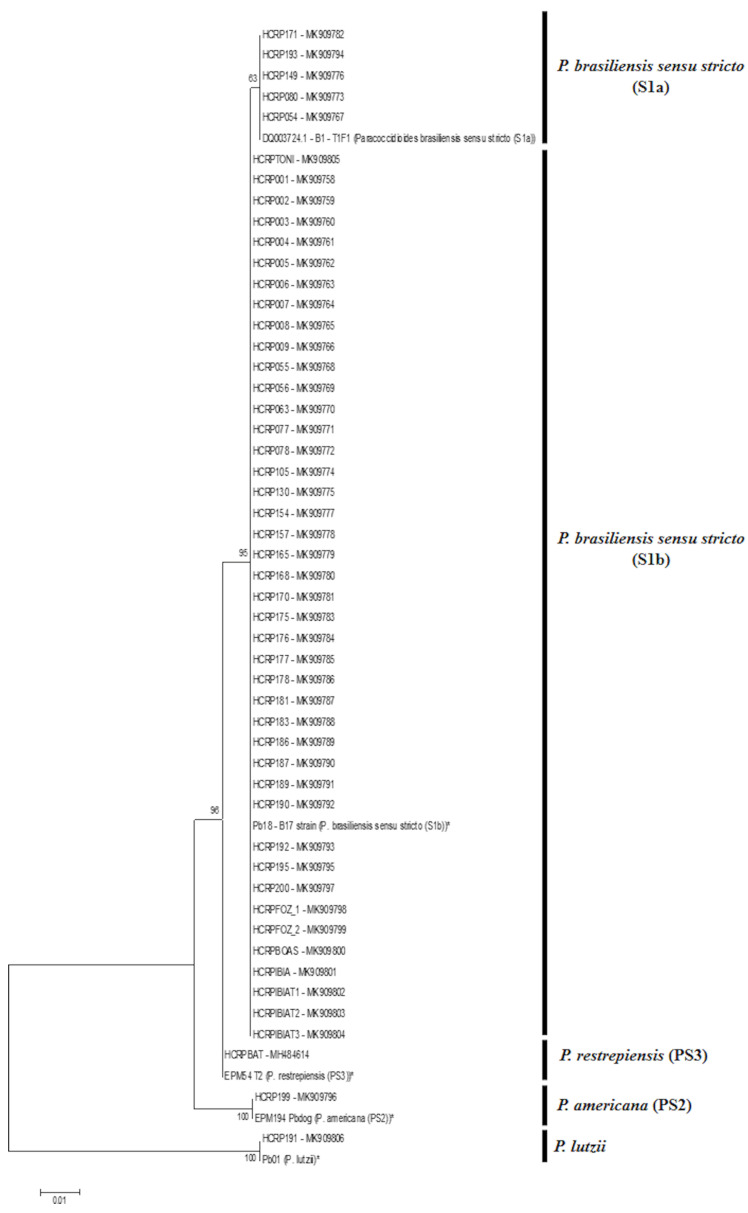
Phylogenetic analysis of *gp43* exon 2 of clinical and environmental isolates of *Paracoccidioides* spp. compared to the reference strains (*), that were also sequenced for this study and had a 100% identity with the deposited sequences on GenBank under accession number: [Pb18—B17—DQ003729.1 (*P. brasiliensis sensu stricto* (S1b); EPM194—Pbdog—DQ003736.1 (*P. americana* (PS2); EPM54—T2—KT251008.1 (*P. restrepiensis* (PS3); Pb01—EU870196.1 (*P. lutzii*)]. Evolutionary history was measured by the maximum likelihood (ML) method based on the Kimura 2 model—parameter model.

**Figure 3 jof-06-00132-f003:**
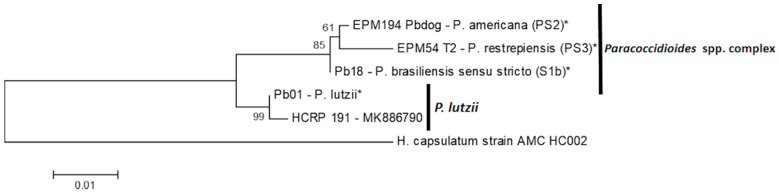
Phylogenetic analysis of the ITS1-5.8S-ITS2 rDNA region of the isolate HCRP191 compared to the reference strains (*), that were also sequenced for this study and had a 100% identity with the deposited sequences on GenBank under accession number: [Pb18—B17 (*P. brasiliensis sensu stricto* (S1b)), EPM194—Pbdog—(*P. americana* (PS2)) and EPM54—T2 (*P. restrepiensis* (PS3))—KT155977.1 for three reference strains; Pb01—EU870297.1 (*P. lutzii*)] and *H. capsulatum* AMC_HC002 are an external group. Evolutionary history was measured by the maximum likelihood (ML) method based on the Tamura–Nei model. The high similarity with Pb01 identifies HCRP 191 as *P. lutzii*.

**Figure 4 jof-06-00132-f004:**
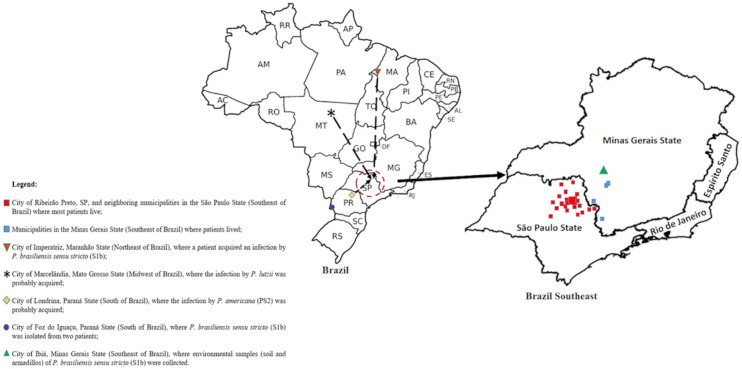
Map of Brazil highlighting the states of São Paulo and Minas Gerais and showing the geographic area centered by Ribeirão Preto, SP (
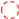
), where most patients lived; the migration of some patients who acquired the infection by *Paracoccidioides* spp. in other regions (

); and the collection site of the environmental isolates (
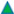
).

**Table 1 jof-06-00132-t001:** Paracoccidoidomycosis clinical form, patient residence and Brazilian region where the infection probably occurred (or collection area in the case of environmental isolates), according to *Paracoccidioides* species.

*Paracoccidoides* Species	Total Number	Clinical Form—*n* (%)		Brazilian State of Residence—*n* (%)				Brazilian Region of *Paracoccidioides* spp. Infection—*n* (%)			
		AC/Subacute	Chronic	SP	MG	PR	MA	SE	S	NE	CW
*P. brasiliensis sensu stricto* (S1b)	38	13(34.2)	25(65.8)	31(81.6)	4(10.5)	2(5.3)	1(2.6)	35(92.1)	2(5.3)	1(2.6)	-
*P. brasiliensis sensu stricto* (S1a)	5	2(40)	3(60)	5(100)	-	-	-	5(100)	-	-	-
*P. americana* (PS2)	1	-	1(100)	1(100)	-	-	-	-	1(100)	-	-
*P. restrepiensis* (PS3)	1	1(100)	-	1(100)	-	-	-	1(100)	-	-	-
*P. lutzii*	1	-	1(100)	1(100)	-	-	-	-	-	-	1(100)
*P. brasiliensis sensu stricto* (S1b) *	4	-	-		4(100)			4(100)	-	-	-

SP—São Paulo State; MG—Minas Gerais State; PR—Paraná State; MA—Maranhão State; SE—Southeast; S—South; NE—Northeast; CW—Center West. (*): Environmental isolates.

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
