# Peer review of "Phylogenetic Species of Paracoccidioides spp. Isolated from Clinical and Environmental Samples in a Hyperendemic Area of Paracoccidioidomycosis in Southeastern Brazil"

_jof, 2020, doi:10.3390/jof6030132_

Round 1

Reviewer 1 Report

Dear authors,

The article presents an interesting work about paracoccidioidomycosis distribution in Brazil. The article is well designed and structured with clear presentation of results and discussion.

Only some point have been fixed to improve readability and understanding of the manuscript, mainly related with figure quality and size. The size of the figures should be increase because is difficult to see the results, also, suplementary figure have to stay out of the text. In the figure 2, the complete number of the type strain and accesion number should be include, and increase the size. In the figure 3, accesion numbers should be included in the phylogenetic tree.

Please check the legend of the figure 4 and remove o change position of the green triangle in the second line.

In line 320, eliminate "XXX"

Best regards

Author Response

The authors thank all suggestion to improve the manuscript.

  1. The size of the Figures was increased.
  2. Type strains numbers and accesion number were added to Figure 2.
  3. Accesion numbers were included in Figure 3.
  4. Legend of figures 2 and 3 have been modified.
  5. Part of the abstract text has been changed. 
  6. The caption in figure 4 was formatted correctly.

Reviewer 2 Report

The reviewed article describes results of the genotyping of fifty isolates of parasitic fungi Paracoccidioides spp. 46 of those were clinical isolates obtained from the same locality in Brazil, and 4 were isolated from environment (one from a soil sample, and three from infected armadillos). The genotyping was done by RFLP of the PCR fragment of conservative tub1 gene and by sequencing of the gp43 exon 2 locus. Actually, though the study design was straightforward, and the results look convincing and are well-complemented with figures and tables, there is no much novelty. Indeed, most of the isolates appeared to belong to two variants of P. brasiliensis, and three other species were represented by single isolates. Neither intraspecies polymorphism, nor new variants or species were detected in the study. 

I would say that the abstract promises much more than the reader may get from the article. Maybe the simple genotyping is still a new approach to link quickly the Paracoccidioides species with the clinical manifestations or epidemiological data of this fungal infection. However, no specific correlation was determined between the species and the disease symptoms. Also having developed molecular phylogenetics for the genus (Teixeira et al. 2009), and genome diversity data for Paracoccidioides species (Munoz et al. 2016) the genotyping performed looks not too sophisticated. 

I wonder why the authors did not make any attempt to sequence the tub1 gene for at least some isolates studied. Especially important it would be in case of the isolate HCRP 191 which sequence was not digested, and which was later identified by ITS sequencing as P. lutzii. The phylogenetic tree on Fig. 3, by the way, looks interesting, and comparison of the ITS loci could provide better inter- and intraspecific resolution than we see in the tree inferred from the gp43 exon 2 locus (Fig. 2). 

I also have some doubts if the work described in the manuscript is really relevant to biogeography of Paracoccidioides spp. Biogeographic studies provide reliable results if they are applied either to free-living and not migrating or migrating along the known routes organisms, or to parasites or symbionts if their hosts are not migrating or migrate using known patterns. When it comes to parasites of humans, who migrate widely, unpredictably and, in the end, randomly, it becomes very difficult to evaluate biogeographic patterns of such parasites. In my opinion, it is not sufficient to use only 4 environmental strains and 46 strains isolated from humans (even with known life history for some of them) to reveal the tendencies of geographic distribution. Also when 47 of 50 isolates belong to P. brasiliensis and do not demonstrate much polymorphism. 

Minor comments:

  • It is important to explain the origin of environmental samples in Materials and Methods section, but not in Supplementary table. Then these samples appear only in Discussion, and up to that moment it remains unclear even what the authors call "environmental"; 
  • There is some problem with Figure 4, at least in my pdf file;
  • the header of subsection 2.4 should be changed into a header, as currently it does not make sense.

Author Response

The authors acknowledge the manuscript revision and suggestions.

General comments

Molecular methods employed in the investigation are simple and well recognized, permitting similar studies by other authors. The original results obtained certainly amplify the knowledge on Paracocccidioides species geographic distribution. The study was not planned to discover new variants or species of Paracoccidoides spp., but to achieve the species that infected paracoccidioidomycosis (PCM) patients in the specific area of Ribeirão Preto, SP, Brazil, where Hospital das Clínicas de Ribeirão Preto Medical School have been a reference center for that patients in the last decades. The P. brasiliensis sensu stricto predominance confirmed data of studies in other São Paulo state geographic area, but isolation of P. restrepiensis, P. americana and P. lutzii was unexpected. With respect to the two last species the authors investigated the respective patient epidemiology and cautiously concluded that fungal infection had origin in other endemic areas. Thus, combining genotypic and epidemiological data it is possible  to conclude about Paracoccidioides spp. (local) biogeography. The authors of the study sended to J. Fungi had favourable conditions to investigate regional Paracoccidioides biogeography considering that fungal strains isolation and genotyping are performed in the same universitary institution where the patients received medical assistance.  By the same motive and remembering that only a specific geographical area was evaluated, the authors consider that 46 clinical isolates and 4 enviromenmental strains (100% of environmental isolations in that area) are sufficient to show the regional prevalence of Paracoccidioides species. For this focal geographic study the authors selected to sequence the gp43 exon 2 loci and do not needed tub1 gene sequencing to distinguish among Paracoccidioides spp.  In respect to a possible association between species and PCM clinical form this study enhances the concept that host factors are more importante to determine the disease manifestation in patients infected by P. brasiliensis  complex.

Minor comments

  1. The origin of the environmental species was added to Materials and Methods section.
  2. Figure 4 was formatted again.
  3. The header of subsection 2.4 was changed as suggest by reviewer 2.
  4. Legend of figures 2 and 3 have been modified.
  5. Part of the abstract text has been changed.

Round 2

Reviewer 2 Report

I am satisfied with the authors' rebuttal for the first review. Now the revised text looks much better not leaving space for doubts and not promising more than the authors really managed to find out. The abstract now also looks much better.

Please check some minor issues with the English.

Lines 65, 144, 216 - there is no need to put a dash within a numeral (forty four but not forty-four)

Line 148 - it is not the isolate which is recognized by two enzymes, but the PCR fragment respective to that isolate

Lines 150-152 - please rewrite the sentences, as they do not make sense both grammatically and biologically.

Line 212 - 60% would be enough (no need to put 60.0%)

Table 1 title - please change the current title for "Paracoccidoidomycosis clinical form, patients residence and Brazilian region where the infection probably occurred (or collection area in case of environmental isolates) according to Paracoccidioides species."

Figure 4 legend. Please fill the brackets.

Author Response

The authors are grateful for the reviewer's suggestions.

All suggestions made by the reviewer were accepted and are highlighted in the text in red.